# Antithrombotic therapy for secondary prevention in patients with stroke or transient ischemic attack: A multiple treatment network meta-analysis of randomized controlled trials

**Dániel Tornyos** [1]☯*, **András Komócsi** [1]☯*, **Alexandra Bálint** [1]‡, **Péter Kupó** [1]‡, **Oumaima El Alaoui El Abdallaoui** [1]‡, **László Szapáry** [2]‡, **László Botond Szapáry** [1]‡

1 Heart Institute, Medical School, University of Pécs, Pécs, Hungary, 2 Department of Neurology, Medical School, University of Pécs, Pécs, Hungary

☯ These authors contributed equally to this work.
‡ AB, PK, OAA, LS and LBS also contributed equally to this work.
* tornyosdaniel@gmail.com (DT); komocsi.andras@pte.hu (AK)

**Data Availability Statement:** The authors had full access to all the data in the study and take

## Abstract

### Objective

As stroke represents one of the leading causes of mortality and disability worldwide, we aimed to determine the preventive effect of different antiplatelet therapies after an ischemic stroke or transient ischemic attack.

### Methods

Network meta-analysis evaluating antiplatelet regimes after an ischemic stroke or transient ischemic attack. Searches were conducted in MEDLINE, EMBASE, and Cochrane Library databases until Nov. 23, 2021, for randomized controlled trials. Direct comparisons within trials were combined with indirect evidence from other trials by using a frequentist model. An additive network meta-analysis model was used to evaluate the influence of individual components. The primary efficacy endpoint was a recurrent stroke, the main safety outcomes were the risk of major bleeding and mortality at the longest available follow-up.

### Results

58 randomized controlled trials (175,730 patients) were analyzed. The analysis involved 20 antithrombotic strategies including different antiplatelet agents, combinations with aspirin, and anticoagulant therapies. Cilostazol proved to be the most efficacious in reducing stroke recurrence and the risk of bleeding (RR = 0.66, 95%CI = 0.55–0.80 and RR = 0.39, 95%CI = 0.08–2.01) compared to aspirin, respectively. Intensification with combinations of aspirin with ticagrelor or clopidogrel resulted in a lower risk of stroke recurrence (RR = 0.79, 95%CI = 0.67–0.93 and RR = 0.79, 95%CI = 0.72–0.87) but carried a higher bleeding risk (RR = 3.01, 95%CI = 1.65–5.49 and RR = 1.78 95%CI = 1.49–2.13).

responsibility for the integrity of the data and the accuracy of the data analysis. In our meta-analysis, all of our baseline data were obtained from the manuscripts and supplementary materials of the included studies. The citations of the included trials can be found in our supplementary material, all the baseline data can be accessed and retrieved from the cited studies. We have also uploaded our study's minimal underlying data set as a private version to Figshare. https://figshare.com/s/731d3ea06f7a636353c1.

**Funding:** This work was supported by the GINOP-2.3.3-15-2016-00031 grant of the Hungarian Government. The financial support does not affect the submitted work, and the researchers are independent from the funder. The funders had no role in study design, data collection and analysis, decision to publish, or preparation of the manuscript.

**Competing interests:** All authors have completed the ICMJE uniform disclosure form at www.icmje.org/coi_disclosure.pdf and declare: no support from any organization for the submitted work; A. KOMÓCSI has received lecture fees from Bayer Healthcare Pharmaceuticals, Eli Lilly, KRKA, MSD, Pfizer, Boehringer-Ingelheim, and Abbot Vascular; no other relationships or activities that could appear to have influenced the submitted work. This does not alter our adherence to PLOS ONE policies on sharing data and materials. For this work, we did not receive any funding from commercial sources.

**Abbreviations:** ACS, acute coronary syndrome; AMP, adenosine monophosphate; ASA, acetylsalicylic acid, aspirin; CI, confidence interval; IS, ischemic stroke; ISTH, International Society on Thrombosis and Hemostasis; MA, meta-analysis; MACE, major cardiovascular adverse events; NMA, network meta-analysis; RCT, randomized controlled trials; RR, risk ratio; TIA, transient ischemic attack; TIMI, The Thrombolysis in Myocardial Infarction.

## Conclusion

The prognosis of patients with an ischemic stroke or transient ischemic attack is improved with antiplatelets. Cilostazol showed the best risk-benefit characteristics without trade-off with the risk of major bleeding. Improved stroke recurrence with intensified antiplatelet regimens is counterbalanced with higher bleeding risk, and consequently, mortality remains unaffected. Treatment decisions in stroke survivals should integrate the assessment of bleeding risk for better identification of patients with the highest benefit of treatment intensification.

## Systematic review registration

Prospero registration number: CRD42020197143, https://www.crd.york.ac.uk/prospero/display_record.php?RecordID=197143.

## Introduction

Stroke, including ischemic and hemorrhagic events, accounts for roughly 10% of all deaths worldwide. Currently, it is the most common cause of disability and the second leading cause of death [1]. In untreated cases with a recent stroke or transient ischemic attack (TIA), the risk of recurrence is the highest. A recurring stroke affects approximately 30% of the cases leading to a deterioration of the neurological symptoms or even death [2, 3].

The current guidelines support the use of antiplatelet therapy in patients with non-cardioembolic ischemic stroke (IS) or TIA [1, 4]. The use of aspirin (ASA) is backed by the most robust clinical evidence; however, the risk of recurrent stroke remains high in ASA-treated patients [3, 5]. Several randomized controlled trials (RCTs) with more potent antiplatelet agents or combinations blocking multiple platelet activation pathways were performed [3, 6–9]. These strategies seem to be beneficial in terms of alleviating the risk of thrombotic events; however, they may come with the price of increased risk of hemorrhagic events [3, 10].

The injured brain parenchyma and fragile cerebral vasculature after IS predispose post-stroke cases to a higher risk of cerebral bleeding. Although a certain risk for bleeding may be acceptable, in this case, hemorrhagic transformation can be fatal. Thus, finding an optimal balance between bleeding and ischemia represents a delicate task.

Alternatives to ASA consist of inhibitors of platelet aggregation and degranulation using several pathways including cyclooxygenase, phosphodiesterase inhibitors, or serotonin receptor antagonists. Platelet ADP $P2Y_{12}$ receptor antagonists applied alone or in combination showed heterogeneous results in stroke studies [3, 11–13].

In this current era where multiple comparisons were tested in clinical studies, with often contradictory results, a comprehensive meta-analysis (MA) investigating a large amount of relevant literature and therapeutic choices can be exceptionally important.

Therefore, we performed an extensive systematic review with network meta-analysis (NMA) to compare and rank the alternative antiplatelet strategies in secondary prevention of stroke. As multiple studies tested different combinations of antiplatelet agents, we aimed to compare their efficacy and safety and to analyze the prevention of the ischemic and hemorrhagic events attributable to the individual components.

## Materials and methods

The review protocol was registered in the PROSPERO database (registration number: CRD42020197143).

Electronic databases including PubMed (MEDLINE), EMBASE, and Cochrane Library (from inception until Nov. 23, 2021) were searched using context-relevant keywords. Articles reporting RCTs of antiplatelet agents in patients with IS or TIA were identified. No language restriction was applied. The queries included the following medical subject heading (MeSH): "Stroke"[MeSH] OR "Ischemic Attack, Transient"[MeSH] AND "Platelet Aggregation Inhibitors"[MeSH] OR "Anticoagulants"[MeSH]. Furthermore, we extended our search with the reference list of relevant articles as well as the identified reviews and guidelines. Fig 1 depicts the process of the literature screening.

In the analysis, we included trials that fulfilled the following criteria: (A) RCTs including patients with IS or TIA with the assessment of clinical efficacy and safety of an antiplatelet-based strategy. (B) Use of an approved antiplatelet in monotherapy or combination, as well as a control group using antiplatelet, anticoagulant treatment, or placebo. (C) Reporting the frequency of recurrent stroke during the follow-up compliant with the intention-to-treat analysis that allows identification of results attributable to the medications. Observational studies including registries and uncontrolled, or cohort studies were excluded.

In a reference manager service (myendnoteweb.com, Clarivate Analytics, USA) records were combined. Duplicates were removed with the identification of similarities of titles, abstracts, authors, and publication years. The explored sources were screened by title, abstract, and full text against our pre-defined eligibility criteria. The full-text screening was performed by two investigators in duplicate without blinding (DT and AB). Discrepancies were resolved with third-party adjudication (AK).

The data extraction included patient and trial characteristics including the criteria of inclusion and exclusion, the nature and applied doses of antiplatelets, length of the treatment period, and follow-up. Details are summarized in S1 and S2 Tables in S1 File.

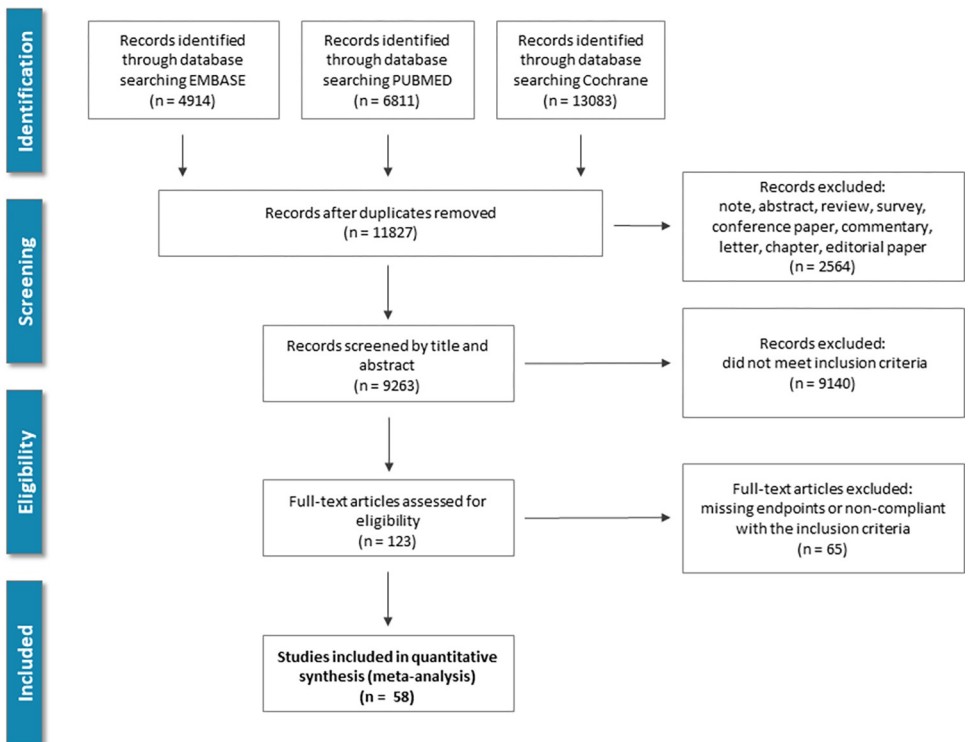

**Fig 1. Study screening and selection flow: Overview of study screening and selection process according to PRISMA guidelines.**

The frequency of recurrent stroke was defined as the primary efficacy endpoint. Mortality and frequency of major bleeding complications were defined as major safety endpoints. Secondary endpoints included the composite of major cardiovascular adverse events (MACE), and its components, (cardiovascular mortality, stroke, and myocardial infarction), disability, IS, hemorrhagic stroke, as well as the rate of different bleeding categories: major and minor bleeding, any relevant bleeding, and intracranial bleeding. The internal definitions of the studies were used to ascertain clinical events. To resolve the use of different major bleeding definitions, we extracted The Thrombolysis in Myocardial Infarction (TIMI) major bleeding [14] and International Society on Thrombosis and Hemostasis (ISTH) major bleeding [15, 16] if available. The endpoints of interest were collected until the longest follow-up available.

Assessment of bias was performed using the Cochrane risk-of-bias and GRADEpro tool for RCTs [17, 18]. Visual asymmetry estimation of a funnel plot supplemented with Egger's linear regression test was performed to assess publication bias (S1 Fig in S1 File).

The risk of the endpoint events was analyzed in a hierarchical NMA using the *netmeta* package (ver. 1.2–1) of the statistical software R (ver. 4.0.0) [19–22]. This package allows estimating NMA models within a frequentist framework, with its statistical approach derived from graph theoretical methods developed for electrical networks [23]. Some medications in a NMA may have common components or be combinations. An additive NMA model assuming the sum of the effects of the individual components was used in the assessment of treatment combinations and their components. Pooled treatment effects for binary endpoints were compared using risk ratio (RR) with 95% confidence intervals (95%CI). The significance of the pooled logRR was determined by the Z-test. $P<0.05$ was considered statistically significant. Statistical inference was based on the results of random-effects model analyses. In the latter, the DerSimonian-Laird $tau^2$ estimator was used to estimate the variance of the distribution of true effect sizes and account for inter-study variability. The choice of the random-effects model was made based on the consideration that the true preventive effect of antiplatelet treatment may vary from study to study influenced by the heterogeneity of the included trials. Furthermore, the random-effects model accounts better for inter-study differences and results in wider credible intervals providing more conservative and robust results.

Inconsistency among studies was quantified utilizing $I^2$. Cochrane Q heterogeneity test ($Chi^2$) was used to assess within and between-treatment-arm-design variation. We present the percent of the $I^2$ together with the p-value of the $Chi^2$ test. The former describes the percentage of total variation across studies that is due to heterogeneity rather than chance. Values of $I^2$ less than 25% were considered as low and $I^2$ greater than 75% as high.

The ranking of treatment was performed using P-scores. P-scores reflect the certainty that one treatment protocol is better than another strategy, averaged over all competing therapies, a method that is equivalent to the surface under the cumulative ranking curve (SUCRA) [20].

## Results

11,827 records were identified by the search and full-text 123 potentially eligible articles were retrieved (Fig 1). Overall, 58 RCTs involving 175,730 (range: 66–21,106) patients done between 1969 and 2021. The trials evaluated 10 antiplatelets as well as 6 antiplatelet combinations. The main characteristics of the included trials and their citations can be found in the supplementary appendix (S1 Table in S1 File). Control groups included alternative antiplatelet therapies, placebo, or one of 4 anticoagulants. The procedural data and clinical characteristics of the patient populations are described in S2 Table in S1 File. The mean study sample size was 1053(66–21106) participants. In total, 140,617 participants were randomly assigned to antiplatelet therapy, 8,757 to anticoagulant treatment, and 26,356 to placebo. The mean age of

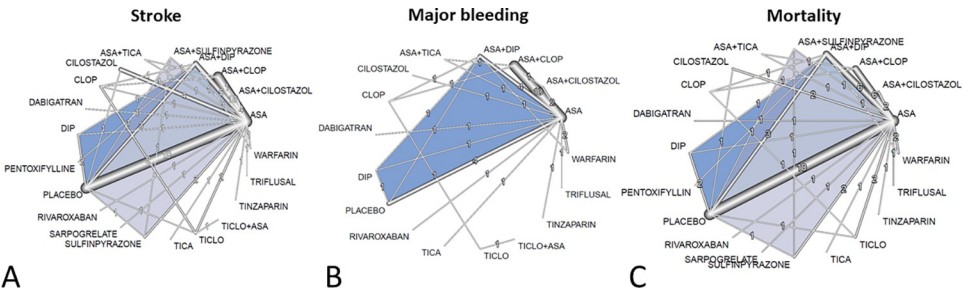

**Fig 2. Network layout and the results of the primary endpoints: Network graphs represent the overall composition of evaluations of the primary endpoints in our network.** The numbers and the width of the edges related to the number of studies within a particular comparison were tested. The blue triangles mark the multi-arm trials. Abbreviations: ASA: aspirin; DIP: dipyridamole; CLOP: clopidogrel; TICA: ticagrelor; TICLO: ticlopidine.

participants was 64±3.3; 58,618 (33.3%) of the sample population were women. The mean duration of the treatment was 480(7–1460) days. Two studies randomly assigned participants to 4 and one to 3 groups, and 14 of 58 were placebo-controlled trials. Consistent with the study protocol, the primary analysis was based on the 58 studies that used drugs within the licensed dose range (i.e., the dosage approved by the regulatory agencies in the USA, Europe, Japan, and/or Korea). Fig 2 shows the network of eligible comparisons for efficacy and safety. The antiplatelet combinations included ASA plus ticlopidine in 1, ASA plus clopidogrel in 13, ASA plus ticagrelor in 2, ASA plus cilostazol in 5, ASA plus sulfinpyrazone in 1, and ASA plus dipyridamole in 10 studies. Each antiplatelet treatment had at least one ASA-controlled trial, except for pentoxifylline and the combination of ASA plus ticlopidine. Analysis of bias showed high quality of the source information with a low probability of possible bias. No obvious publication bias was found (S1, S2/A, and S2/B Figs in S1 File). In the studies comparing cilostazol vs. aspirin the stroke, mortality, and bleeding outcomes, according to the GRADEpro evaluation, reflected in moderate certainty. For the cilostazol plus aspirin vs. aspirin trials, the assessment found low certainty, because the total number of patients involved in these 4 studies was small, and the number of the observed endpoints was low. The summary of this analysis is added as S3/A and S3/B Table in S1 File.

The netleague table in the appendix provides detailed results of pairwise MA (S4 Table in S1 File). Analyses of the network showed consistency within as well as between designs. Design-specific decomposition showed heterogeneity with ASA plus ticlopidine and ASA plus dipyridamole versus ASA; however, after detaching of single designs showed homogenous results (S5 Table in S1 File). Similarly, netheat analyses showed consistency regarding the effect estimates of direct and indirect comparisons in the network (S3 Fig in S1 File).

## Results for the primary outcomes

Fig 3 illustrates the network meta-analysis' results for the primary outcomes. In terms of efficacy cilostazol, clopidogrel, and combinations of ASA with clopidogrel, dipyridamole, and ticagrelor were more effective than ASA, with relative risks ranging between RR = 0.66, 95% CI = 0.55–0.80 for cilostazol and RR = 0.85, 95%CI = 0.77–0.94 for clopidogrel. The combinations of ASA plus ticagrelor, cilostazol, and clopidogrel had similar efficacy as RR = 0.79. Placebo treatment significantly increased the risk of stroke (RR = 1.21, 95%CI = 1.12–1.30).

In terms of major bleeding (35 RCTs, comprising 144,088 patients), the risk was higher with warfarin, rivaroxaban, and ASA plus clopidogrel, while highest with ASA plus ticagrelor combination (RR = 3.01, 95%CI = 1.65–5.49). Compared to ASA significantly lower risk was

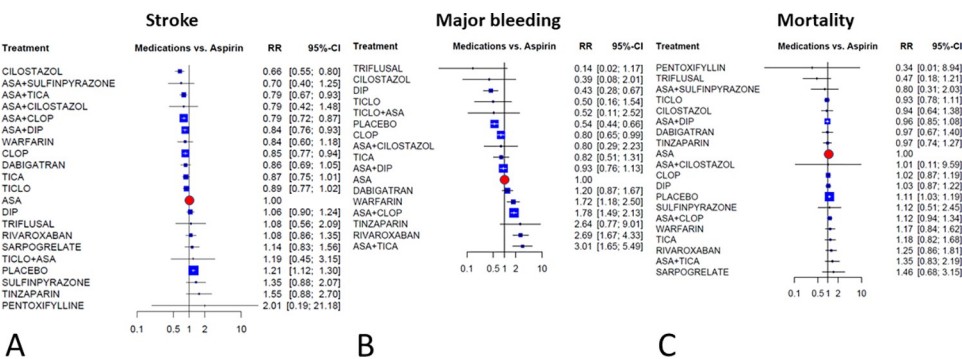

**Fig 3. Results of the primary endpoints: Forest plots present the findings of the random-effects network meta-analyses as contrasting aspirin monotherapy set as reference.** Abbreviations: ASA: aspirin; DIP: dipyridamole; CLOP: clopidogrel; TICA: ticagrelor; TICLO: ticlopidine; RR: risk ratio; CI: confidence interval.

seen with placebo, clopidogrel, and dipyridamole. While the lowest relative risk was seen with triflusal (RR = 0.14, 95%CI = 0.02–1.17) and cilostazol (RR = 0.39, 95%CI = 0.08–2.01).

Estimates of mortality (46 RCTs, comprising 159,788 patients) did not show significant differences compared to ASA, except for placebo which was associated with a significant 11% increase in mortality.

## Results for disability secondary outcome

Twenty-two RCTs reported disability outcomes in form of Rankin scale (14 RCTs), deterioration using SNSS (1 RCT), or an increase in NIHSS scale (3 RCTs). Network analysis of these trials showed improved outcomes with ASA plus cilostazol, and with dabigatran compared to ASA; however, these benefits do not reach the level of significance. S4/A and S4/B Fig in S1 File depict the analysis of disability outcomes.

## Ranking of treatment strategies

For stroke prevention cilostazol ranked the highest followed by combination therapies with ASA plus sulfinpyrazone, clopidogrel, ticagrelor, and cilostazol, respectively. Contrasting its stroke prevention potential ASA plus ticagrelor ranked the last when major bleeding was considered, and triflusal, dipyridamole, and cilostazol ranked the highest. ASA ranked in the middle third among the tested antiplatelet strategies with regards to both stroke prevention and major bleeding. Importantly, all antiplatelet strategies received higher P-values than ASA in at least one of the two aspects. (Fig 4).

## Effect of the individual antiplatelets in the component network meta-analysis models

The component analysis reflected that among antiplatelets the use of ASA, cilostazol, dipyridamole, and the P2Y$_{12}$ blocker clopidogrel, ticlopidine, and ticagrelor conveyed a significant reduction of stroke risk. With cilostazol showing the most effective risk reduction (RR = 0.60) the RRs fall in a similar range among the other drugs (RR = 0.78–0.88). The risk of bleeding was significantly increased with ASA, clopidogrel, and ticagrelor treatment. All these three antiplatelets lifted bleeding risk similarly (RR = 1.78–2.23) (Fig 5).

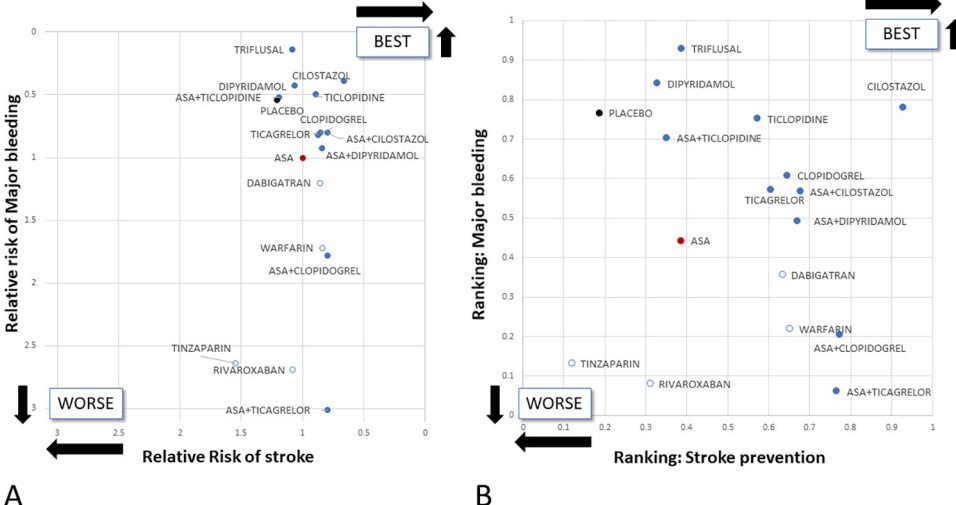

**Fig 4.** Cluster analysis of the different medication strategies: Panel A depicts the risk of stroke and major bleeding relative to aspirin monotherapy. Panel B shows the treatment ranking regarding these two aspects. Placebo is marked with black, aspirin with red, antiplatelet treatments with blue, and anticoagulants with white dots. Abbreviation: ASA: aspirin.

## Stratified analysis according to the severity of stroke and the pharmacological and the follow-up protocol

In order to determine the possible impact of differences in the pharmacological interventions, length of follow-up and to explore how far the severity of the stroke affects the results, several stratified analyses were performed. The analysis of the available trials stratified according to the stroke severity based on the trials inclusion criteria showed consistent results with the full model only with minor differences. (S6 Table in S1 File).

Furthermore, studies were labeled as 'acute' if patients were enrolled within 48 hours of the stroke event and 'chronic' if this criterion was not met. According to the follow-up length, further subgroups were defined with follow-up limited to one month or three months, or unlimited. In this analysis in addition to the efficacy of dipyridamole, which showed a comparable difference (full model: RR = 0.87, 95%CI = 0.79–0.95; acute studies: RR = 0.49, 95%CI = 0.2–1.22; chronic studies: RR = 0.9, 95%CI = 0.8–1), only minor disproportions were found when comparing the acute and chronic care protocols (S7 Table in S1 File).

The subgroup of antiplatelet studies was defined with the exclusion of trials testing anticoagulant medication, and this stratification did not show any disparity (S7 Table in S1 File).

These stratified analyses strengthen our results and show that the most effective therapy is cilostazol in terms of ischemic and bleeding events or even mortality as the results were consistent in the investigated clinical strata.

## Discussion

### Principal findings

This comprehensive systematic review included 58 parallel-group RCTs of 175,730 patients with IS or TIA. The review compared 14 single and 6 ASA combined active antithrombotic interventions or placebo in a single framework. The risk of clinical endpoints including the risk of stroke and bleeding events were assessed to inform the decision-making with a ranking among the therapies.

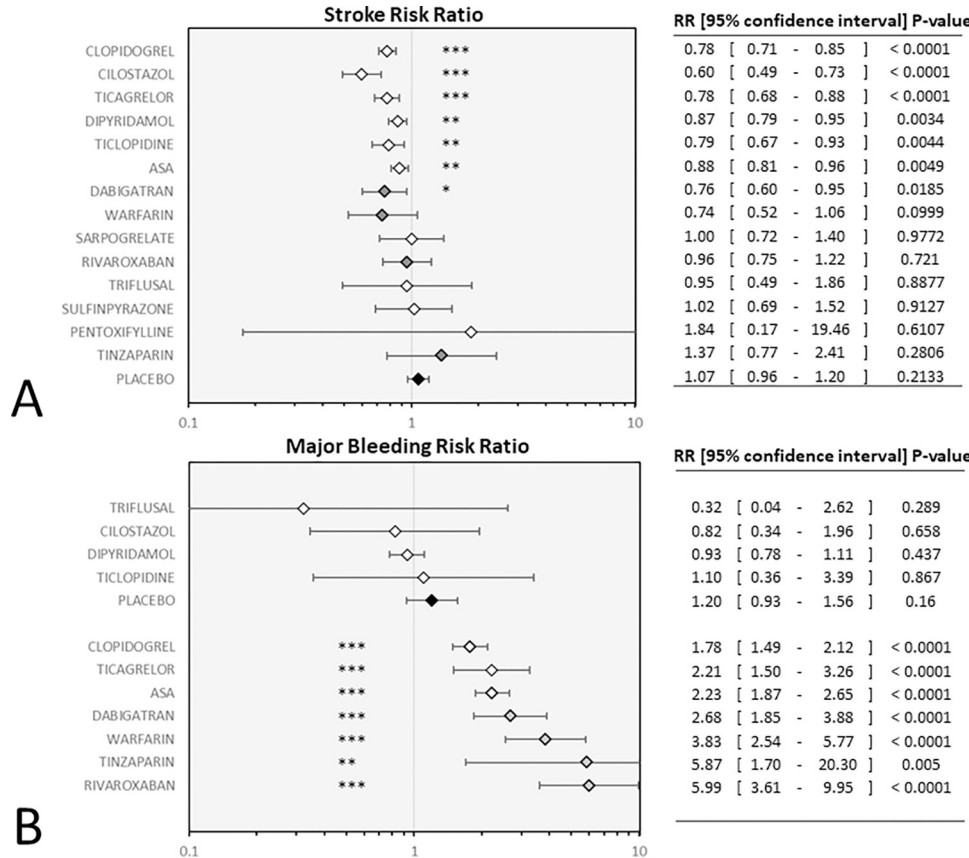

**Fig 5. Individual antiplatelet drugs in the component network meta-analysis model: Forest plots depict the risk reduction of stroke (panel A) and the risk of major bleeding (panel B) associated with the individual treatment components.** White diamonds mark the relative risk attributable to the addition of antiplatelet treatment. Gray diamonds mark anticoagulants black diamond marks placebo control. Abbreviation: ASA: aspirin.

The results of this NMA suggest that among single and dual antiplatelet therapies cilostazol has the most favorable risk-benefit balance for secondary prevention of IS or TIA. Cilostazol resulted in the highest reduction of the stroke recurrence risk (34%) alongside a numerical 61% reduction in the risk of bleeding compared to ASA. The complex mechanism of action of cilostazol may explain the outstanding results reflected in the secondary prevention of IS and the risk of bleeding which makes it an unmatched and advantageous choice among the anti-platelet drugs. Cilostazol is a selective inhibitor of phosphodiesterase-3 (PDE- 3), that prevents the inactivation of the platelet intracellular second messenger the cyclic adenosine monophosphate (AMP) impeding platelet aggregation [24, 25]. Further potential mechanisms include depletion of intracellular calcium together with an endogenous nitric oxide pathway enhanced vasodilation [25]. Some data suggest potential effects that delay the onset of atherosclerosis [26], including reducing the proliferation of arterial smooth muscle cells and safeguarding the vascular endothelium [27, 28]. Cilostazol also has anti-inflammatory and neuroprotective effects [29–31].

Our findings support the results of earlier analyses where cilostazol proved to be the most preferred choice between antiplatelet therapies in stroke [32]. In addition to this former study, in our NMA we extended it with the inclusion of the most recent trials testing intensified anti-platelet regimes as well as studies comparing antiplatelets to anticoagulants, and other alternative platelet inhibitors.

In cases of acute IS within 24 to 48 hours, administration of antiplatelet monotherapy, ASA is supported by the current American Heart Association/American Stroke Association (AHA/ASA) guidelines. The initiation of the treatment should be postponed until the next day if thrombolysis is administered. In the case of ASA contraindications, the use of other antithrombotic treatments may be considered. As the chances of recurrent stroke are the highest in the acute phase, intensified strategies may be advantageous. Therefore, within 24 hours after minor non-cardioembolic IS the use of dual antiplatelet strategies for up to 21 days is optional. Although these therapeutic choices may be considered, their efficacy and safety have not been fully demonstrated, and further evidence is needed [1, 33].

Importantly, when compared to ASA any of the investigated antiplatelet strategies showed benefits of either a lower risk of bleeding or in terms of more effective stroke prevention. Among monotherapies besides cilostazol, clopidogrel also significantly reduced both bleeding and stroke risk (20% and 15% reduction of RR, respectively).

Dual antiplatelet therapy using ASA with clopidogrel or ticagrelor belongs to the standards of treatment in acute coronary syndromes (ACSs) without or with stent placement [34]. These intensified antiplatelet schemes also more effectively alleviated stroke recurrence. However, this effect of both combinations was counterbalanced with a greater risk of bleeding. Importantly, mortality did not improve with these combinations.

The prodrug thienopyridines including ticlopidine, clopidogrel, and prasugrel, and the direct-acting antagonists such as ticagrelor and cangrelor compose the two main types of P2Y$_{12}$ receptor inhibitors used in different clinical settings. These drugs were extensively evaluated in several fields of preventive cardiology and have an established role in the treatment of coronary disease [34]. However, in neurology, their benefits are less well established. Only preliminary data are available with cangrelor in IS cases requiring stenting, while prasugrel increased the rate of intracranial and fatal bleeding in ACS patients with a history of cerebrovascular events [35]. On the contrary, an important body of evidence reflects that dual antiplatelet therapy composed of ASA and clopidogrel may improve the recurrence of stroke. The direct comparisons of clopidogrel and ASA dual antiplatelet therapy with ASA were assessed in 10 trials consisting of one of the most robust edges in our network analysis. This strategy resulted in a 21% reduction in stroke but also a 78% increase in major bleeding risk. Furthermore, ticagrelor when combined with ASA, assessed based on the indirect evidence and the recently published THALES trial, effectively prevented stroke recurrence. This latter trial demonstrated a 21% reduction of recurrent stroke compared to ASA. However, the bleeding risk also tremendously increased by 201% [3].

It is important to highlight that following cilostazol monotherapy the combinations of ASA with ticagrelor, clopidogrel, and dipyridamole showed the best results in reducing the risk of recurrent stroke. Contrasting these combinations cilostazol plus ASA showed an advantage in both endpoints compared to ASA, while a higher risk of bleeding can be expected for the former combinations.

## Limitations

Our analysis should be interpreted considering some limitations. The severity of stroke showed some variations within the included trials. Most of the studies recruited cases with minor stroke or a high-risk TIA (according to the National Institutes of Health Stroke Scale (NIHSS) score of 3 or less (range, 0 to 42, with higher scores indicating more severe stroke), or a score of 4 or higher on the ABCD$^2$ scale (range, 0 to 7, with higher scores indicating a higher risk of stroke), respectively) [10]. The current diagnostic algorithms may be insufficient to exclude in some cases a potential cardioembolic origin. Oral anticoagulation remains the

treatment of choice for these patients, and thus we focused our review of antiplatelet strategies on trials where patients with cardioembolic sources were excluded. All analyses were performed by pooling the active drug arms with various dosages and treatment duration; therefore, it limits our ability to assess how the differential effects of the dosage of these drugs affect the outcomes. Moreover, capturing bleeding events was also hampered by the paucity of one overall accepted bleeding definition system.

Our meta-analysis reflected that cilostazol is a promising choice for secondary prevention of stroke, supported by its lower risk of ischemic and hemorrhagic complications. However, research testing cilostazol included predominantly Asian population while data regarding other ethnic groups are limited. The incidence of complications may differ among ethnic groups with a higher risk of hemorrhagic stroke in the Asian population, and with a more frequent occurrence of myocardial infarction and ischemic complications among Caucasians [36]. We included 10 RCTs determining the effect of cilostazol in the secondary prevention of ischemic stroke. Despite the advantages shown in the chronic phase, in most of these trials cilostazol showed no increased benefits over aspirin in the acute phase. However, it must be noted that in these studies patients were included at different times after the index event. It is necessary to highlight that in the CAIST trial cilostazol showed lower rates of recurrent stroke in the acute phase [37]. This result is consistent with our stratified analysis according to the inclusion and follow-up protocol of the involved trials. On the other hand, due to the small number of acute-phase studies; moreover, as stroke etiologies and other contributing factors may also differ, more evidence and large-scale RCTs are needed to confirm our results.

## Conclusion

Stroke still represents an important target for the improvement of medical therapy being the second leading cause of mortality and the most common ground for disability. In our NMA, we summarized the available clinical evidence of antiplatelet treatments in the secondary prevention of IS or TIA. Comprising a wide set of trials, the analysis supports strong evidence in favor of cilostazol and clopidogrel regarding secondary prevention of IS. Compared to ASA these drugs in monotherapy effectively reduced the risk of recurrent stroke without an increase in the risk of bleeding. Among the active dual antiplatelet therapies, ASA plus ticagrelor, clopidogrel, or dipyridamole were the most efficacious treatment regimens to prevent recurrent stroke. However, in cases of these intensified schemes, a tradeoff was seen with an increased risk of major bleeding. We hope that our comprehensive NMA would help to inform the recommendations and guide clinical decisions to choose the most appropriate therapy for secondary prevention of IS.

## Supporting information

**S1 Checklist. PRISMA 2020 checklist.**
(DOCX)

**S1 File. The online supplemental material contains the following: S1, S2 A/B, S3, S4 A/B Figs, S1, S2, S3 A/B, S4-S7 Tables, and the citations of the included trials.** The legends of the included materials can be found in the supplemental material file.
(DOCX)

## Author Contributions

**Conceptualization:** Dániel Tornyos, András Komócsi.

**Data curation:** Dániel Tornyos, Alexandra Bálint.

**Formal analysis:** Dániel Tornyos, András Komócsi.

**Funding acquisition:** Dániel Tornyos, András Komócsi, Alexandra Bálint.

**Investigation:** Dániel Tornyos.

**Methodology:** Dániel Tornyos, András Komócsi.

**Project administration:** Dániel Tornyos, András Komócsi.

**Resources:** Dániel Tornyos, András Komócsi.

**Software:** Dániel Tornyos, András Komócsi.

**Supervision:** András Komócsi.

**Validation:** Dániel Tornyos, András Komócsi.

**Visualization:** Dániel Tornyos, András Komócsi.

**Writing – original draft:** Dániel Tornyos, András Komócsi.

**Writing – review & editing:** Dániel Tornyos, András Komócsi, Alexandra Bálint, Péter Kupó, Oumaima El Alaoui El Abdallaoui, László Szapáry, László Botond Szapáry.

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
