## [Decision Letter · Decision Letter 0]

19 May 2022

PONE-D-21-34982

Antithrombotic therapy for secondary prevention in patients with stroke or transient ischemic attack: a multiple treatment network meta-analysis of randomized controlled trials

PLOS ONE

Dear Dr. Tornys,

Thank you for submitting your manuscript to PLOS ONE. After careful consideration, we feel that it has merit but does not fully meet PLOS ONE’s publication criteria as it currently stands. Therefore, we invite you to submit a revised version of the manuscript that addresses the points raised during the review process.

The manuscript does not reach to an enough level for acceptance to the journal in the present form. See the Reviewers' comments and respond them appropriately.

We look forward to receiving your revised manuscript.

Kind regards,

Masaki Mogi

Academic Editor

PLOS ONE

Journal Requirements:

2. Thank you for stating the following in the Competing Interests section: "All authors have completed the ICMJE uniform disclosure form at www.icmje.org/coi_disclosure.pdf and declare: no support from any organization for the submitted work; A. KOMÓCSI has received lecture fees from Bayer Healthcare Pharmaceuticals, Eli Lilly, KRKA, MSD, Pfizer, Boehringer-Ingelheim, and Abbot Vascular; no other relationships or activities that could appear to have influenced the submitted work."

We note that you received funding from a commercial sources: Bayer Healthcare Pharmaceuticals, Eli Lilly, KRKA, MSD, Pfizer, Boehringer-Ingelheim, and Abbot Vascular

4. Please upload a new copy of Figures 1, 2, 3, and 5 as the detail is not clear. Please follow the link for more information: https://blogs.plos.org/plos/2019/06/looking-good-tips-for-creating-your-plos-figures-graphics/" https://blogs.plos.org/plos/2019/06/looking-good-tips-for-creating-your-plos-figures-graphics/

7. Thank you for submitting the above manuscript to PLOS ONE. During our internal evaluation of the manuscript, we found significant text overlap between your submission and the following previously published works, some of which you are an author.

- https://pubmed.ncbi.nlm.nih.gov/34063551/

Please revise the manuscript to rephrase the duplicated text, cite your sources, and provide details as to how the current manuscript advances on previous work. Please note that further consideration is dependent on the submission of a manuscript that addresses these concerns about the overlap in text with published work.

Reviewers' comments:

Reviewer's Responses to Questions

**Comments to the Author**

1. Is the manuscript technically sound, and do the data support the conclusions?

Reviewer #1: Yes

Reviewer #2: Yes

2. Has the statistical analysis been performed appropriately and rigorously? 

Reviewer #1: Yes

Reviewer #2: Yes

3. Have the authors made all data underlying the findings in their manuscript fully available?

Reviewer #1: Yes

Reviewer #2: Yes

4. Is the manuscript presented in an intelligible fashion and written in standard English?

Reviewer #1: Yes

Reviewer #2: Yes

5. Review Comments to the Author

Reviewer #1: Tornyos et al performed a network meta-analysis of 58 RCTs that evaluated the efficacy of different antiplatelet therapies in the secondary prevention of ischemic stroke. Searches were performed in Medline, Embase and Cochrane Library databases through November 2021 for RCTs. The primary endpoint was recurrent stroke and the primary safety endpoints were risk of major bleeding and mortality at longest available follow up. The authors found that cilostazol had the best risk-benefit characteristics without increased risk of major bleeding.

It is noted that cilostazol, in acute studies and acute studies with less than 90 days follow up, showed no benefit with respect to recurrent stroke compared to ASA. The authors note that the risk of recurrent stroke is highest in the month following the index event, and yet cilostazol therapy did not show a reduction in recurrent stroke in this population when compared to ASA. While in the overall analysis of pooled results, cilostazol showed a significant reduction in recurrent stroke compared to ASA monotherapy, it failed to show benefit in the acute setting, presumably first 30 days, following index event. Thus the reduction seen in recurrent stroke compared to ASA in the overall analysis is driven by the reduction in the “chronic” setting, which has a broad definition in this NMA. Can the authors comment in the discussion/limitations regarding this important finding?

Reviewer #2: I read the manuscript entitled "Antithrombotic therapy for secondary prevention in patients with stroke or transient ischemic attack: a multiple treatment network meta-analysis of randomized controlled trials" and I find it interesting and well written. Congratulations for the nice work. The methodology is solid and the conclusions drawn are clinically relevant, although they differ from the common practice that has considered monotherapy with ASA the first line of treatment in the secondary prevention of stroke for years. I particularly appreciated the sub-analyzes shown in Tables 5 and 6, which stratify data according to important elements such as the degree of severity, the pharmacological and the follow-up protocol of the included trials.

I have some minor comments:

- In the methods, you assert that the effect of treatment combinations is the sum of the effects of its components, but it not necessarily true.

- Considering a 30-day follow-up, the association ASA + clopidogrel shows greater efficacy (RR 0.55), as I think it was intended to demonstrate with the sub-analyzes of CHANCE and POINT to establish the optimal duration of DAPT around 21 days. However, the follow-up should not be confused with the duration of the therapy: in fact, in the CHANCE the DAPT lasts only 21 days, then the treatment group continues with only clopidogrel until the 90th day. This wording in the tables with the studies characteristics should be corrected.

- Studies testing cilostazol have another substantial problem besides ethnicity: sample size. In fact, out of 10 RCTs, 4 are well under 400 subjects and two slightly exceed this number, generating a certain degree of "imprecision" which reduces the quality of the evidence, according to the Grade Pro. It might be useful to conclude the work with a Summary of Findings table according to the Grade Pro to evaluate the quality of the evidence obtained.

- In my opinion, some data on disability is also missing because it is one of the main effects of stroke that significantly impacts the patient's quality of life, but I know that not all the studies considered have evaluated this outcome.

6. PLOS authors have the option to publish the peer review history of their article (what does this mean?). If published, this will include your full peer review and any attached files.

Reviewer #1: No

Reviewer #2: No

---

## [Author Response · Author response to Decision Letter 0]

30 Jun 2022

PONE-D-21-34982

Dear Editor-in-Chief,

Thank you very much for your mail regarding our manuscript referenced as PONE-D-21-34982.

Enclosed we submit the revised version of our manuscript entitled "Antithrombotic therapy for secondary prevention in patients with stroke or transient ischemic attack: a multiple treatment network meta-analysis of randomized controlled trials".

We really do appreciate the comments of the reviewers that we consider to be essential in order to improve the comprehensiveness of the article. According to this, we did our best to rewrite and correct each and every outstanding point. Please refer to the attached corrected manuscript as well as to the Track Changes version for the detailed modifications.

Sincerely yours,

Dániel Tornyos MD

András Komócsi MD DSc

Medical School, University of Pécs 

Reviewers’ comments:

Reviewer #1:

Tornyos et al performed a network meta-analysis of 58 RCTs that evaluated the efficacy of different antiplatelet therapies in the secondary prevention of ischemic stroke. Searches were performed in Medline, Embase and Cochrane Library databases through November 2021 for RCTs. The primary endpoint was recurrent stroke and the primary safety endpoints were risk of major bleeding and mortality at longest available follow up. The authors found that cilostazol had the best risk-benefit characteristics without increased risk of major bleeding.

It is noted that cilostazol, in acute studies and acute studies with less than 90 days follow up, showed no benefit with respect to recurrent stroke compared to ASA. The authors note that the risk of recurrent stroke is highest in the month following the index event, and yet cilostazol therapy did not show a reduction in recurrent stroke in this population when compared to ASA. While in the overall analysis of pooled results, cilostazol showed a significant reduction in recurrent stroke compared to ASA monotherapy, it failed to show benefit in the acute setting, presumably first 30 days, following index event. Thus the reduction seen in recurrent stroke compared to ASA in the overall analysis is driven by the reduction in the “chronic” setting, which has a broad definition in this NMA. Can the authors comment in the discussion/limitations regarding this important finding?

Response:

We appreciate the comment, and we have updated our limitations accordingly. “We included 10 RCTs determining the effect of cilostazol in the secondary prevention of ischemic stroke. Despite the advantages shown in the chronic phase, in most of these trials cilostazol showed no increased benefits over aspirin in the acute phase. However, it must be noted that in these studies patients were included at different times after the index event. It is necessary to highlight that in the CAIST trial cilostazol showed lower rates of recurrent stroke in the acute phase [37]. This result is consistent with our stratified analysis according to the inclusion and follow-up protocol of the involved trials. On the other hand, due to the small number of acute-phase studies; moreover, as stroke etiologies and other contributing factors may also differ, more evidence and large-scale RCTs are needed to confirm our results.”

Reviewer #2:

I read the manuscript entitled "Antithrombotic therapy for secondary prevention in patients with stroke or transient ischemic attack: a multiple treatment network meta-analysis of randomized controlled trials" and I find it interesting and well written. Congratulations for the nice work. The methodology is solid and the conclusions drawn are clinically relevant, although they differ from the common practice that has considered monotherapy with ASA the first line of treatment in the secondary prevention of stroke for years. I particularly appreciated the sub-analyzes shown in Tables 5 and 6, which stratify data according to important elements such as the degree of severity, the pharmacological and the follow-up protocol of the included trials.

I have some minor comments:

1. In the methods, you assert that the effect of treatment combinations is the sum of the effects of its components, but it not necessarily true.

Response:

Thank you very much for your comment. Indeed, the summary effect is not present in all studies exactly, however, when choosing an appropriate modeling method for statistical analysis not always an exact behavior should be considered. The combination therapies have more than zero order influence on one another, and the order of the summation is not reaching a level of second or third order thus we considered that the use of a simple summation model would be appropriate.

2. Considering a 30-day follow-up, the association ASA + clopidogrel shows greater efficacy (RR 0.55), as I think it was intended to demonstrate with the sub-analyzes of CHANCE and POINT to establish the optimal duration of DAPT around 21 days. However, the follow-up should not be confused with the duration of the therapy: in fact, in the CHANCE the DAPT lasts only 21 days, then the treatment group continues with only clopidogrel until the 90th day. This wording in the tables with the studies characteristics should be corrected.

Response:

The supplementary table was corrected accordingly.

3. Studies testing cilostazol have another substantial problem besides ethnicity: sample size. In fact, out of 10 RCTs, 4 are well under 400 subjects and two slightly exceed this number, generating a certain degree of "imprecision" which reduces the quality of the evidence, according to the Grade Pro. It might be useful to conclude the work with a Summary of Findings table according to the Grade Pro to evaluate the quality of the evidence obtained.

Response:

We agree with the reviewer's remark. The design of cilostazol studies affected consistency and precision. In the studies comparing cilostazol vs. aspirin the stroke, mortality, and bleeding outcomes, according to the GRADEpro evaluation, reflected in moderate certainty. For the cilostazol plus aspirin vs. aspirin trials, the assessment found low certainty, because the total number of patients involved in these 4 studies was small, and the number of the observed endpoints was low. The summary of this analysis is added as S3/A and S3/B Tables.

4. In my opinion, some data on disability is also missing because it is one of the main effects of stroke that significantly impacts the patient's quality of life, but I know that not all the studies considered have evaluated this outcome.

Response:

We really welcome your comment as disability plays a significant role after an IS. Twenty-two RCTs reported disability outcomes in form of Rankin scale (14 RCTs), deterioration using SNSS (1 RCT), or an increase in NIHSS scale (3 RCTs). Network analysis of these trials showed improved outcomes with ASA plus cilostazol, and with dabigatran compared to ASA; however, these benefits do not reach the level of significance. S4/A and S4/B Figures depict the analysis of disability outcomes.

---

## [Decision Letter · Decision Letter 1]

3 Aug 2022

Antithrombotic therapy for secondary prevention in patients with stroke or transient ischemic attack: a multiple treatment network meta-analysis of randomized controlled trials

PONE-D-21-34982R1

Dear Dr. Tornyos,

We’re pleased to inform you that your manuscript has been judged scientifically suitable for publication and will be formally accepted for publication once it meets all outstanding technical requirements.

Kind regards,

Masaki Mogi

Academic Editor

PLOS ONE

Additional Editor Comments (optional):

No further comment.

Reviewers' comments:

Reviewer's Responses to Questions

**Comments to the Author**

1. If the authors have adequately addressed your comments raised in a previous round of review and you feel that this manuscript is now acceptable for publication, you may indicate that here to bypass the “Comments to the Author” section, enter your conflict of interest statement in the “Confidential to Editor” section, and submit your "Accept" recommendation.

Reviewer #1: All comments have been addressed

Reviewer #2: All comments have been addressed

2. Is the manuscript technically sound, and do the data support the conclusions?

Reviewer #1: Yes

Reviewer #2: Yes

3. Has the statistical analysis been performed appropriately and rigorously? 

Reviewer #1: Yes

Reviewer #2: Yes

4. Have the authors made all data underlying the findings in their manuscript fully available?

Reviewer #1: Yes

Reviewer #2: Yes

5. Is the manuscript presented in an intelligible fashion and written in standard English?

Reviewer #1: Yes

Reviewer #2: Yes

6. Review Comments to the Author

Reviewer #1: (No Response)

Reviewer #2: (No Response)

7. PLOS authors have the option to publish the peer review history of their article (what does this mean?). If published, this will include your full peer review and any attached files.

Reviewer #1: No

Reviewer #2: No

---

## [Editor Report · Acceptance letter]

8 Aug 2022

PONE-D-21-34982R1 

Antithrombotic therapy for secondary prevention in patients with stroke or transient ischemic attack: a multiple treatment network meta-analysis of randomized controlled trials 

Dear Dr. Tornyos:

I'm pleased to inform you that your manuscript has been deemed suitable for publication in PLOS ONE. Congratulations! Your manuscript is now with our production department. 

Kind regards, 

on behalf of

Dr. Masaki Mogi 

Academic Editor

PLOS ONE